# Using Additives for Fouling Control in a Lab-Scale MBR; Comparing the Anti-Fouling Potential of Coagulants, PAC and Bio-Film Carriers

**DOI:** 10.3390/membranes10030042

**Published:** 2020-03-12

**Authors:** Petros Gkotsis, Anastasios Zouboulis, Manassis Mitrakas

**Affiliations:** 1Laboratory of Chemical and Environmental Technology, Department of Chemistry, Faculty of Sciences, Aristotle University of Thessaloniki, 54124 Thessaloniki, Greece; petgk@chem.auth.gr (P.G.); zoubouli@chem.auth.gr (A.Z.); 2Analytic Chemistry Laboratory, Department of Chemical Engineering, School of Engineering, Aristotle University of Thessaloniki, 54124 Thessaloniki, Greece

**Keywords:** membrane bio-reactors, membrane fouling, biomass additives, coagulants/flocculants, powdered activated carbon, bio-film carriers

## Abstract

This study investigates the effect of different additives, such as coagulants/flocculants, adsorption agents (powdered activated carbon, PAC), and bio-film carriers, on the fouling propensity of a lab-scale membrane bio-reactor (MBR) treating synthetic municipal wastewater. The coagulation agents FO 4350 SSH, Adifloc KD 451, and PAC1 A9-M at concentrations of 10 mg/L, 10 mg/L, and 100 mg Al/L, respectively, and PAC at a concentration of 3.6 ± 0.1 g/L, exhibited the best results during their batch-mode addition to biomass samples. The optimal additives FO 4350 SSH and Adifloc KD 451 were continuously added to the bioreactor at continuous-flow addition experiments and resulted in increased membrane lifetime by 16% and 13%, respectively, suggesting that the decrease of SMP_c_ concentration and the increase of sludge filterability is the dominant fouling reduction mechanism. On the contrary, fouling reduction was low when PAC1 A9-M and PAC were continuously added, as the membrane lifetime was increased by approximately 6%. Interestingly, the addition of bio-film carriers (at filling ratios of 40%, 50%, and 60%) did not affect SMP_c_ concentration, sludge filterability, and trans-membrane pressure (TMP). Finally, the effluent quality was satisfactory in terms of organics and ammonia removal, as chemical oxygen demand (COD), biochemical oxygen demand (BOD)_5_, and NH${}_{4}^{+}$-N concentrations were consistently below the permissible discharge limits and rarely exceeded 30, 15, and 0.9 mg/L, respectively.

## 1. Introduction

The membrane bio-reactor (MBR) is a state-of-the-art technology that combines the activated sludge process with membrane filtration for the treatment of municipal and industrial wastewaters. Membrane bioreactors offer a series of advantages, such as high quality effluent, removal of pathogens, and diminishing the utilization of chemicals for disinfection, allowing it to be regarded as an environmentally friendly technology for wastewater treatment. However, membrane fouling still remains the major drawback in MBR systems preventing their universal application over the last few decades [1]. Fouling is attributed to a variety of components present in the (waste) water, which increase the membrane’s resistance either by their adsorption or deposition onto its surface, or even by complete pore-blocking. Fouling leads to permeate flux decline, which in turn decreases the time intervals necessary for membrane cleaning and replacement, resulting in both higher capital and operating costs. According to the reversibility of flux, fouling can be (i) reversible, when it is removed by the application of physical cleaning methods; (ii) irreversible, when it is removed by the application of chemical cleaning methods; or (iii) irrecoverable, when it cannot be removed by the application of a physical or chemical cleaning method owing to the long-term use of the membrane after a series of consecutive cleaning cycles.

According to the nature of foulants, that is, the substances that cause membrane fouling, fouling can be also divided into (i) bio-fouling, owing to the formation of a bio-film layer, caused mainly by the presence of microorganisms that are attached and growing on the membrane surface; (ii) organic fouling, which is caused by organic compounds, such as polysaccharides, proteins, humic substances, and other organic bio-polymers; and (iii) inorganic fouling, which refers to the deposition of inorganic materials, like salts, metal oxides, and so on [2]. Among the various compounds that exist in the activated sludge, the extracellular polymeric substances (EPSs) are considered to be the most significant foulants during the operation of MBR systems. EPSs include a variety of organic macromolecules, usually polysaccharides and proteins, which are located outside the bacteria cells or inside the microbial aggregates. They are classified into bound EPS (bEPS) or soluble EPS (sEPS), which are also known as soluble microbial products (SMPs). In particular, the carbohydrate fraction of SMP (SMP_c_) has been often cited as the most significant constituent causing fouling in MBRs, although the role of protein compounds in fouling has not been yet fully elucidated [3].

Over the last few years, several methods/techniques have been employed in order to control and/or mitigate membrane fouling in MBRs. These methods can be distinguished into conventional and innovative methods. The conventional methods have been implemented since the earlier years of MBR technology and include the operation of these systems under low fluxes; the application of physical methods, such as aeration with coarse bubbles, backwashing, and relaxation; as well as the application of chemical methods by utilization of simple chemical reagents solutions such as of NaOCl, citric acid, or oxalic acid. The innovative methods are implemented more recently and include the application of electric field, ultrasound, ozone, and various membrane surface modifications. It must be stated that, in most cases, the innovative methods are not applied individually, but they are usually implemented in combination with a conventional method, aiming to result in more effective reduction of fouling [4,5]. Regarding fouling characterization and quantification, various direct or indirect methods have been implemented, which include either standardized tests, such as the Delft filtration, the drainage test, the time-to-filer (TTF) method, and so on, or the measurement of specific foulants concentration, such as colloidal total organic carbon (TOC), SMP, and so on [4,6].

One of the most promising strategies for fouling control is the use of appropriate additives in the MBR, which alter the biomass characteristics, resulting in the improvement of the filtration process and increase of the membrane lifetime. The most commonly applied additives usually include coagulation/flocculation agents, adsorbents, and bio-film carriers (BFCs) or bio-carriers [7,8,9,10]. Dosing of coagulants into MBR systems has been reported to mitigate fouling owing to the reduction of SMP in the supernatant liquor and to the formation of large flocs, which limit the blockage of membranes pores [11]. The coagulation agents, which are used for fouling control in MBRs, usually fall into three main categories: inorganic monomers, inorganic polymers, and organic polymers. The simultaneous adsorption and bio-degradation, rather than a single biological process, reflect the major advantage of using adsorbents in MBRs [12]. Powdered activated carbon (PAC) is one of the most widely applied adsorption agents. Its addition to MBRs mitigates fouling by improving the removal of low molecular weight organics and EPS and SMP, which are considered to be primarily responsible for membrane fouling. PAC can also decrease the compressibility of sludge flocs and increase the porosity of cake layer, resulting in the increase of membrane flux [9,13,14]. Bio-film carriers (BFCs) or bio-carriers are plastic or sponge-made materials that have been also reported to mitigate fouling by direct physical scouring of membrane and by reducing the amount of SMPs [4].

As shown above, coagulants/flocculants, PAC, and bio-carriers have been extensively used in MBRs for fouling control and/or mitigation. However, most studies concern batch-mode addition of these materials, making it difficult to assess the of fouling reduction in full-scale MBR applications where long-term operation is applied. In this study, the effect of coagulants, PAC, and bio-carriers on the membrane fouling of a lab-scale MBR is investigated and their performance during batch-mode and continuous-flow addition is compared. Furthermore, the present study is aimed at the development of a low-cost and time-saving methodology, through the evaluation of optimal additives based on the modification of the SMP_c_ concentration and filterability tests. To the author’s best knowledge, this is the first study that systematically investigates and compares the anti-fouling potential of the most significant additive categories (at various dosages), including 14 commonly applied coagulation agents, PAC, and bio-film carriers.

## 2. Materials and Methods 

### 2.1. Lab-Scale MBR Operation and Additives

The lab-scale MBR consisted of three main sub-units (Figure 1a): (i) wastewater feed unit (200 L), (ii) bioreactor with submerged membrane (20 L), and (iii) permeate collection unit (40 L), and its operation was as follows (Figure 1b): synthetic municipal wastewater (Table 1), with a composition based on Organization for Economic Co-operation and Development (OECD) guidelines [15], was fed as the substrate for the biological treatment process. The feed was led by a peristaltic pump to the aeration tank (bioreactor), where the concentration of the dissolved oxygen (DO) was monitored by a DO-meter in the range of 2–3 mg/L. The air needed for the biomass, as well as for the cleaning of the applied membrane, was supplied by an air compressor, the pressure of which was appropriately reduced to the desired value by means of an air pressure reducer/valve. Gas and liquid flow meters were applied for the measurement of aeration and feed or permeate flow rates, respectively, while the level of the mixed liquor in the membrane/aeration tank was controlled by level sensors. Another peristaltic pump was used to withdraw the permeate from the upper end of the membrane and a high-resolution pressure transmitter was employed in order to continuously record the evolution of trans-membrane pressure (TMP). The permeate collection unit was the final recipient of the produced permeate. A flat sheet, microfiltration membrane (Kubota Membranes Inc., Osaka, Japan) with a pore size of 0.4 μm and an effective area of 0.11 m^2^ (made of chlorinated polyethylene) was used, while one-minute relaxation steps were performed every nine minutes of operation. This membrane type is commonly applied in MBR systems, as it is known to provide high rejection rates, owing to the gradual formation of a cake layer on the membrane surface that acts as a ‘secondary membrane’ and enhances the filtration process. It should also be stressed that the operation of the lab-scale MBR system was fully automated: a programmable logic controller (PLC) (Simatic S7-1200, SIEMENS, Nuremberg, Germany) was employed to control the operation of the peristaltic pumps, DO-meter, level sensors, and pressure transducer. Following the initial inoculation of the bioreactor with activated sludge, which was received from the municipal wastewater treatment plant of Thessaloniki (Northern Greece), the system was operated continuously in order to achieve steady-state conditions.

First, the batch-mode addition of 14 commercially available (commonly used) coagulation agents (Table 2) and PAC (size > 100 μm, density: 0.4–0.5 g/cm^3^, ash content ≤ 10%, humidity ≤ 4%) was conducted at different concentrations in appropriately received biomass samples. Bio-film carriers were not added to the samples, as the short-term nature of batch experiments would not allow the development of biomass onto their surface, which favors the adsorption of SMP_c_. In order to examine the effect of solids retention time (SRT) on the process, the addition of coagulants and PAC was conducted at three SRTs (10, 15, and 20 days). During this phase (batch-mode addition experiments), which lasted for 384 days, activated sludge was not renewed and the acclimatization period for each SRT was determined as 2 × SRT, because this is generally considered to be an adequate amount of time for the adjustment of biomass to new conditions. The appropriate amount of sludge (2, 1.33, and 1 L) was daily withdrawn from the bioreactor (20 L), depending on the desired SRT (10, 15, and 20 days, respectively). The concentration of mixed liquor suspended solids (MLSS) in the bioreactor was 6.1 ± 1.1, 8.0 ± 1, and 9.2 ± 0.8 g/L, while organic loading, expressed by the ratio F/M (food to microorganisms), was approximately 0.20, 0.15, and 0.13 g chemical oxygen demand (COD)/g MLSS∙d, for SRT = 10, 15, and 20 days, respectively.

The additives that exhibited the best results during the batch-mode experiments were then continuously added to the bioreactor. Addition of optimal coagulants and PAC concentrations was conducted according to the following mass balance equation (Equation (1)), previously proposed by Gkotsis et al. [16]:(1)$${\overset{•}{m}}_{\mathrm{in}}\;=\;{\overset{•}{m}}_{\mathrm{acc}}\;+\;{\overset{•}{m}}_{\mathrm{out}}$$
where ${\overset{•}{m}}_{\mathrm{in}}$ is the additive mass flow rate into the bioreactor (kg/h), ${\overset{•}{m}}_{\mathrm{acc}}$ is the additive mass that is accumulated in the aeration tank per time (kg/h), and ${\overset{•}{m}}_{\mathrm{out}}$ is the additive mass flow rate removed from the MBR (kg/h). During this phase (continuous-flow addition experiments), which lasted for 417 days, bio-film carriers were also added to the bioreactor at three filling ratios (40%, 50%, and 60%) and the activated sludge was renewed after the addition of different additives. In addition, the lab-scale MBR operated under constant SRT (20 days), after removing the appropriate amount of sludge (1 L) from the bioreactor (20 L) on a daily basis. The concentration of MLSS in the bioreactor was 10.8 ± 1.5 g/L, while F/M was approximately 0.11 g COD/g MLSS∙d.

It should also be stated that the membrane was chemically cleaned when the TMP reached the maximum TMP limit (20 kPa), however, a new membrane module was implemented when a new SRT was applied during the batch-mode addition or a new additive was tested during the continuous-flow addition.

### 2.2. Fouling Estimation

During both phases, irreversible fouling was assessed in terms of SMP_c_ removal, according to the phenol-sulfuric acid method, and reversible fouling was assessed in terms of sludge filterability tests, according to the standard TTF (time-to-filter) method. Total fouling was estimated during the continuous-flow addition of the optimal additives by measuring the evolution of TMP.

SMPs were extracted by the following procedure: mixed liquor samples (50 mL) were obtained daily from the bioreactor and centrifuged (for 20 min at 2000 rpm) in order to separate the solid biomass. Then, for the determination of the carbohydrate fraction of SMPs in the supernatant, the phenol-sulfuric acid method was employed [17], which is the most widely used colorimetric method for the determination of carbohydrate concentration in aqueous solutions. The principle of this method is that carbohydrates, when dehydrated by reaction with concentrated sulfuric acid, produce furfural derivatives. Further reaction between furfural derivatives and phenol develops a detectible color. A short description of the standard procedure is as follows: 1 mL aliquot of a carbohydrate solution was mixed with 1 mL of wt. 5% aqueous solution of phenol in a test tube. Subsequently, 5 mL of conc. H_2_SO_4_ was added rapidly to the mixture. After allowing the test tubes to stand for 10 min, they were vortexed for 30 s and placed for 20 min in a water bath at 25 °C for color development. Then, light absorption at 480 nm was recorded on a spectrophotometer. Reference solutions were prepared in identical manner as aforementioned, except that the 1 mL aliquot of carbohydrate was replaced by glucose. A Hitachi UV/vis double-beam spectrophotometer was used for these measurements. Irreversible fouling was assessed in terms of the following ratio (Equation (2)):a_SMPc_ = SMP_additive_/SMP_no additive_(2)
where a_SMPc_ is the ratio of SMP_c_ concentration after the addition of an additive (i.e., coagulant or PAC) in the mixed liquor, to the SMP_c_ concentration before this addition (i.e., the respective blank measurement). It is evident that the lower ratio a_SMPc_ indicates that the tested concentration is more effective in terms of SMP_c_ removal.

The time-to-filter (TTF) method is a well-established method that can be used as an easy and relatively rapid way to assess the mixed liquor filterability. During the TTF method, a Buchner funnel with a diameter of 90 mm and Whatman #1 and #2, that is, pore size of 11 μm and 8 μm, respectively, or equivalent filter papers are used (in the present study, Whatman #1 was used). Following its removal from the bioreactor, sludge amount of 200 mL is instantly poured on the Buchner funnel and the time required to obtain 100 mL of filtrate is measured at a vacuum pressure of 51 kPa (designated as TTF). It is understood that low TTF values indicate high biomass filterability, while high TTF values indicate low biomass filterability [18,19]. Reversible fouling was assessed in terms of the following ratio (Equation (3)):b_TTF_ = TTF_additive_/TTF_no additive_(3)

Similarly, b_TTF_ is the ratio of TTF recorded after the addition of an additive in the mixed liquor, to the TTF recorded before this addition, and the lower ratio b_TTF_ indicates that sludge filterability is more enhanced. The sum of two ratios a_SMPc_ + b_TTF_ was also calculated in order to determine the additives that reduce both SMP_c_ concentration and TTF the most and, consequently, are likely to mitigate both irreversible and reversible membrane fouling.

### 2.3. Determination of Effluent Quality Parameters

The quality parameters of the lab-scale MBR effluent (concentrations of COD, total nitrogen (TN), NH_4_^+^-N, and NO_3_^−^-N) were determined with standardized Hack–Lange LCK test kits (with part numbers 314, 238, 304, and 339, respectively), along with a DR-3900 spectrophotometer. Biochemical oxygen demand (BOD)_5_ was measured with a respirometric BOD_5_ system (Oxi700, Orbeco-Hellige, Sarasota, FL, USA). The present study is part of a scientific research project that focuses primarily on the influence of additives on membrane fouling and secondarily on the efficient removal of effluent quality parameters. Therefore, the lab-scale MBR system was not initially designed to include an anaerobic tank for promoting enhanced biological phosphorus removal, and thus phosphorus removal was not examined in the present work (however, a series of construction changes is currently being conducted in order to render the system more flexible, as well as to allow for the efficient phosphorus removal).

## 3. Results and Discussion

### 3.1. Fouling Investigation

#### 3.1.1. Batch-Mode Addition of Coagulants and PAC

During the batch-mode experiments, coagulants were added to mixed liquor samples at concentrations of 5, 7.5, and 10 mg/L for organic polymers and 50, 75, and 100 mg Fe or Al/L for inorganic coagulants, according to preliminary jar tests that were previously conducted in the laboratory (data not presented). PAC was added to mixed liquor samples at 0.5–5 g/L. These concentrations are similar to those employed in relevant literature studies [20,21,22] that investigate membrane fouling mitigation in several applications.

Although ratios a_SMPc_ and b_TTF_ are different parameters, they describe two sides of the same phenomenon (fouling). As stated in Section 2.2, ratio a_SMPc_ describes irreversible fouling in terms of SMP_c_ removal, while ratio b_TTF_ describes reversible fouling in terms of biomass filterability enhancement. However, estimating each additive’s anti-fouling potential based solely on the assessment of each ratio separately might lead to the wrong conclusions regarding the additives’ competence to mitigate total fouling in the bioreactor. In addition, our previous study [16] has shown that this methodology could be misleading, as there are additives that affect only one type of fouling. Therefore, results are presented in terms of the sum a_SMPc_ + b_TTF_.

Figure 2 demonstrates the sum a_SMPc_ + b_TTF_ after the addition of all coagulants in the biomass samples that were received from the bioreactor when it was operated at SRT = 20, 15, or 10 d. As shown, the coagulants that exhibited the lowest sum a_SMPc_ + b_TTF_ are FO 4350 SSH at 10 mg/L, Adifloc KD 451 at 10 mg/L, and PAC1 A9-M at 100 mg Al/L, that is, two cationic (organic) polyelectrolytes and the pre-polymerized (inorganic) coagulant of Al. These coagulants can be characterized as optimal and are likely to contribute to total fouling mitigation, if added to the bioreactor in continuous-flow mode. It must also be stated that the presented sums resulted from the average values of ratios a_SMPc_ and b_TTF_, as each coagulant concentration was tested three times (relative standard deviation (RSD) = 1%–20%).

Determination of the optimal polyelectrolyte dose is fundamental in order to estimate the coagulant cost for effective mitigation of membrane fouling. The optimal dose of 10 mg/L for application of the cationic polyelectrolytes FO 4350 SSH and Adifloc KD 451 results in the consumption of 1 kg polyelectrolyte for each 100 m^3^ of wastewater. As the commercial cost of FO 4350 SSH and Adifloc KD 451 is in the range of 3 ± 0.5 €/kg and the dissolution/pumping energy is in the range of 2 ± 0.5 kWh/kg, with energy cost around 0.1 €/kWh, the utilization of polyelectrolytes increases the treatment cost by 32 ± 5 €/10^3^ m^3^. However, it must be clarified that this treatment cost is an order of magnitude lower than the biological treatment of municipal wastewaters.

Figure 3a shows the sum a_SMPc_ + b_TTF_ after the addition of PAC in the biomass samples at concentrations of 0.5–5 g/L. As shown in Figure 3a, the PAC concentrations that exhibited the lowest sum a_SMPc_ + b_TTF_ were 3, 3.5, and 4 g/L. Aiming to achieve further accuracy, addition of PAC for a shorter concentration range (3.1–3.9 g/L) was conducted (Figure 3b). The results showed that the optimal PAC concentration was 3.6 ± 0.1 g/L. As in the case of coagulants, the presented sums resulted from the average values of the ratios a_SMPc_ and b_TTF_, as each PAC concentration was tested three times (RSD = 1%–12%). In Figure 2 and Figure 3, the effect of SRT on the sum a_SMPc_ + b_TTF_ can also be seen. As shown, for most coagulants and PAC concentrations, the decrease of SMP_c_ and TTF is higher at SRT = 20 d.

#### 3.1.2. Continuous-Flow Addition of Coagulants, PAC, and Bio-Carriers

Following the batch-mode addition experiments, continuous-flow addition of optimal coagulants and PAC concentrations was conducted. During this phase, the lab-scale MBR operated under constant SRT= 20 days and fouling was examined by measuring the reduction of SMP_c_ concentration, the decrease of TTF, as well as the evolution of TMP. Moreover, bio-film carriers were also added to the bioreactor at filling ratios of 40%, 50%, and 60% in order to study their influence on fouling.

As shown in Figure 4a, the continuous-flow addition of optimal coagulants resulted in the reduction of SMP_c_ concentration, especially the addition of cationic polyelectrolytes FO 4350 SSH and Adifloc KD 451. On the contrary, the reduction of SMP_c_ concentration after the addition of PAC or bio-carriers was found to be rather low (Figure 4b,c). The same trend was also observed for the measured TTF values; however, the respective TTF reduction by the optimal coagulation agents was not so high (Figure 5). Interestingly, the differences observed in the SMP_c_ concentrations are depicted in the evolution of TMP, verifying the crucial role of carbohydrates in membrane fouling (Figure 6). Therefore, the continuous-flow addition of polyelectrolytes FO 4350 SSH and Adifloc KD 451 increased the membrane lifetime by 16% and 13%, respectively. It is also noteworthy that the addition of bio-film carriers did not change any of the examined fouling indices (SMP_c_, TTF, and TMP).

The results of batch-mode and continuous-flow addition of coagulants and PAC are compared in Figure 7. It is reminded that low values of ratio a_SMPc_ indicate increased SMP_c_ removal, while low values of ratio b_TTF_ indicate enhanced sludge filterability. As shown in Figure 7a, both the batch-mode and the continuous-flow addition of optimal coagulants can significantly reduce the concentration of SMP_c_ at percentage >50%. However, the reduction is higher when the coagulants are continuously added to the bioreactor. On the contrary, sludge filterability is more improved when the coagulants are added to biomass samples (Figure 7b). When coagulants are added to biomass samples, positive electrical charge is instantly produced, neutralizing the negatively charged biomass, resulting in the improvement of sludge filterability, which is expressed as a reduction of filtration time during the short-term TTF experiments. In contrast, the long-term polymerization process, which takes place during the continuous-flow addition of coagulants, is favorable for the entrapment of SMP_c_, resulting in higher reduction of their concentration in the biomass.

Regarding the addition of PAC, it is observed that PAC reduces the concentration of SMP_c_ and the reduction is higher when it is added to biomass samples (Figure 7c). Similarly, TTF is not significantly reduced when PAC is continuously added to the bioreactor (Figure 7d). The apparent reason for the higher adsorption ability of PAC lies in the short-term nature of the batch-mode addition experiments. Shortly after their addition to biomass samples, PAC particles adsorb SMP_c_ components and reduce their concentration, also resulting in the reduction of TTF. It is understood that this reduction cannot be so evident during the continuous-flow addition of PAC, owing to the competitive adsorption of other organic components at a long residence time.

### 3.2. Effluent Quality

The most significant contaminants demanding removal from municipal wastewaters are the suspended solids, organic matter and ammonia, because most of the total nitrogen encountered in domestic sewage is present in this form [23,24]. In the present study, the concentration of suspended solids in the effluent of the lab-scale MBR was almost zero, as no solids were initially added for the preparation of the synthetic wastewater and owing to the microfiltration treatment of effluent. Therefore, the effluent quality was examined in terms of organics (COD, BOD_5_) and nitrogen removal (TN, NH_4_^+^-N, NO_3_^−^-N).

Figure 8 and Figure 9 show the evolution of organics and nitrogen in the effluent of the lab-scale MBR, while Figure 10 depicts the development of pH, temperature, and DO in the biomass of the bioreactor. In these figures, the period of SRT = 20 days refers to the initial period of the second phase when the optimal coagulants were continuously added. However, the results were similar for all other cases as well (addition of PAC/bio-carriers or no additive addition to the bioreactor), that is, the system’s removal efficiency was unaffected by the use of different additives or the absence of them. Vertical dotted lines mark the acclimatization period for each SRT, which was determined as 2 × SRT. The effluent concentrations of COD and BOD_5_ rarely exceeded 30 mg/L and 15 mg/L, respectively. Moreover, almost complete nitrification took place in the bioreactor, which can be attributed to the constant DO concentration (2.5 ± 0.5 mg/L) (Figure 8). The concentration of NH${}_{4}^{+}$-N in the effluent was very low (<1 mg/L) and the respective removal rate often exceeded 99% (Figure 9a). On the contrary, the absence of anoxic reactor resulted in quite high NO_3_^−^-N concentrations in the effluent (Figure 9b). Nevertheless, although denitrification did not take place, a slight total nitrogen removal (≥25%) was observed, owing to its assimilation by the microorganisms for further biomass production (Figure 9c).

It should also be mentioned that the overall system performance was exceptional: pH, temperature, and DO concentration of the biomass were generally satisfactory and met the needs of microbial metabolism (Figure 10). As shown in Figure 10a, apart from the initial biomass adjustment period, where some abrupt fluctuations are observed, pH varies in the range of 7.3–7.8, while temperature follows the respective seasonal variations (12–31 °C). Similar observations were made by other researchers as well [25]. Finally, the automation of the aeration, which is achieved by the PLC, allowed for the effective control of DO concentration in the biomass, as most values range between 2 and 3 mg/L (Figure 10b). Some excessively high or low DO values are sporadically observed in the timeline owing to the instantaneous adhesion of fine air bubbles onto the DO electrode membrane or to the removal of the DO probe from the bioreactor for calibration purposes.

The investigation of biomass viability was beyond the scope of the present study and, therefore, no toxicity tests were conducted. However, the aforementioned results (Figure 8, Figure 9 and Figure 10) could imply the absence of toxicity in the mixed liquor, indicating that the addition of coagulants, PAC, and bio-carriers did not affect biomass viability as well.

## 4. Conclusions

The present study examines the effect of different additives (coagulations agents, PAC, and bio-film carriers) on membrane fouling of a lab-scale MBR, as well as on COD, BOD_5_, and nitrogen removal and nitrification. Batch-mode experiments included the addition of coagulation agents and PAC to biomass samples, and SMP_c_ and TTF modifications were employed as fouling indices. Continuous-flow experiments included the continuous, inline addition of the optimal coagulation agents and PAC to the bioreactor of the MBR and SMP_c_, TTF, and TMP modifications were employed as fouling indices. Bio-film carriers were also added during this phase. The results showed that the coagulation agents, which effectively reduced both the SMP_c_ concentration and TTF during their batch-mode addition, were FO 4350 SSH at 10 mg/L, PAC1 A9-M at 100 mg Al/L, and Adifloc KD 451 at 10 mg/L. Similarly, the optimal PAC concentration was 3.6 ± 0.1 g/L. Continuous-flow addition of polyelectrolytes FO 4350 SSH and Adifloc KD 451 decreased SMP_c_ concentration and TTF, resulting in the increase of membrane lifetime by 16% and 13%, respectively, as indicated by the evolution of TMP. On the contrary, fouling reduction with PAC addition was very low. It was interesting to notice that the addition of bio-film carriers at filling ratios of 40%, 50%, and 60% did not affect SMP_c_, TTF, and TMP. Finally, concerning the overall operation and effluent quality of MBR, COD, BOD_5_, and NH${}_{4}^{+}$-N concentrations in the effluent were far below the respective legislation limits, rarely exceeding 30 mg/L, 15 mg/L, and 0.9 mg/L, respectively, and were not affected by the continuous-flow addition of coagulation agents and PAC.

## Figures and Tables

**Figure 1 membranes-10-00042-f001:**
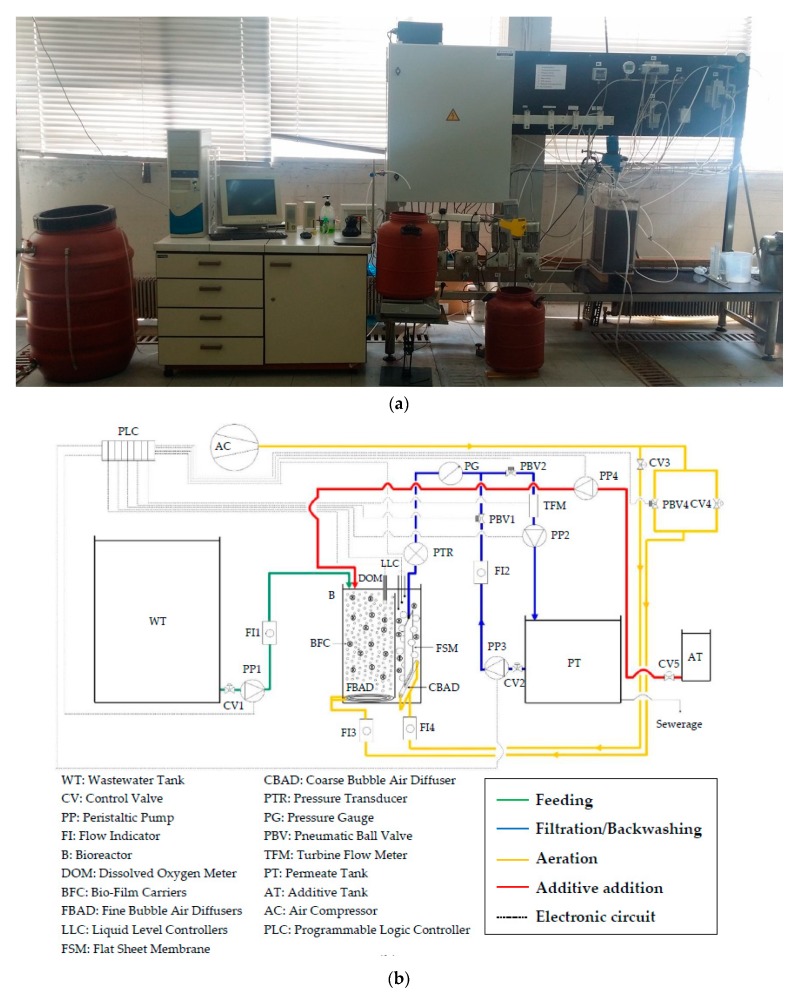
(**a**) Lab-scale membrane bio-reactor (MBR); (**b**) schematic of the lab-scale MBR.

**Figure 2 membranes-10-00042-f002:**
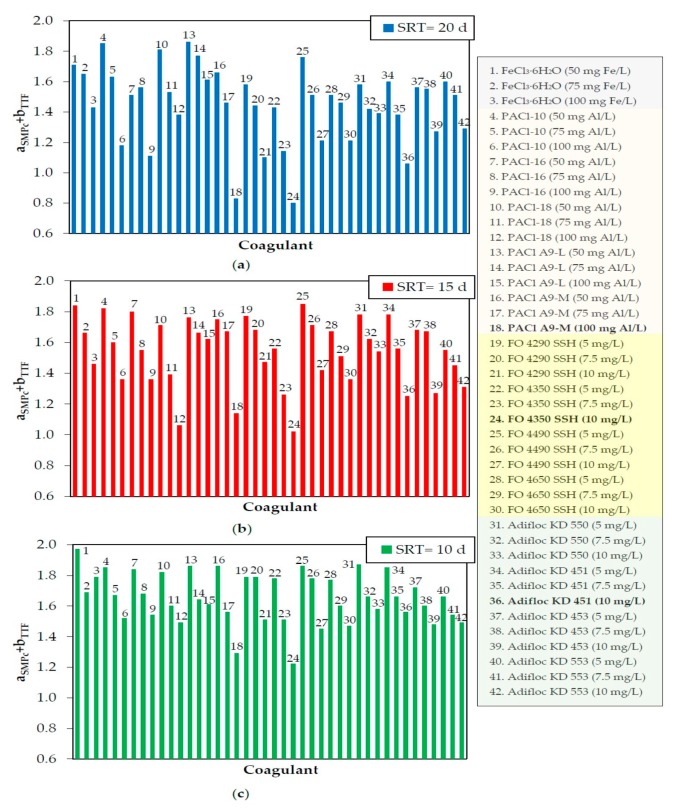
Sum a_SMPc_ + b_TTF_ after the batch-mode addition of coagulants for (**a**) solids retention time (SRT) = 20 d, (**b**) SRT = 15 d, or (**c**) SRT = 10 d. SMP, soluble microbial product; TTF, time-to-filter.

**Figure 3 membranes-10-00042-f003:**
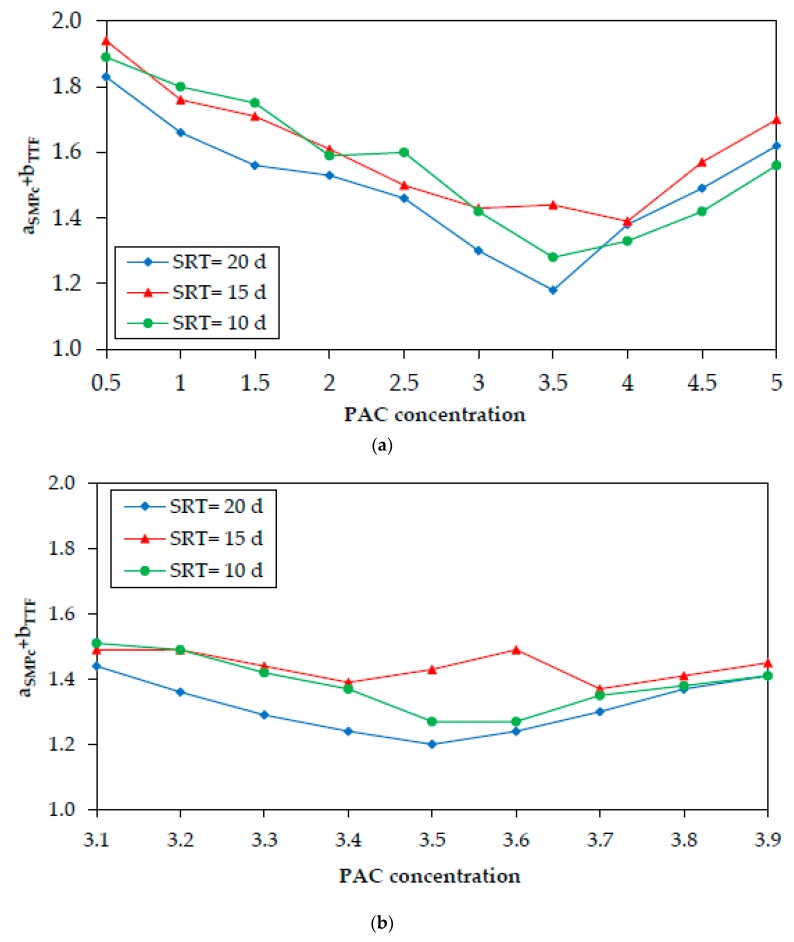
Sum a_SMPc_ + b_TTF_ after the batch-mode addition of powdered activated carbon (PAC) at (**a**) 1–5 g/L and (**b**) 3.1–3.9 g/L, for SRT = 20, 15, or 10 d.

**Figure 4 membranes-10-00042-f004:**
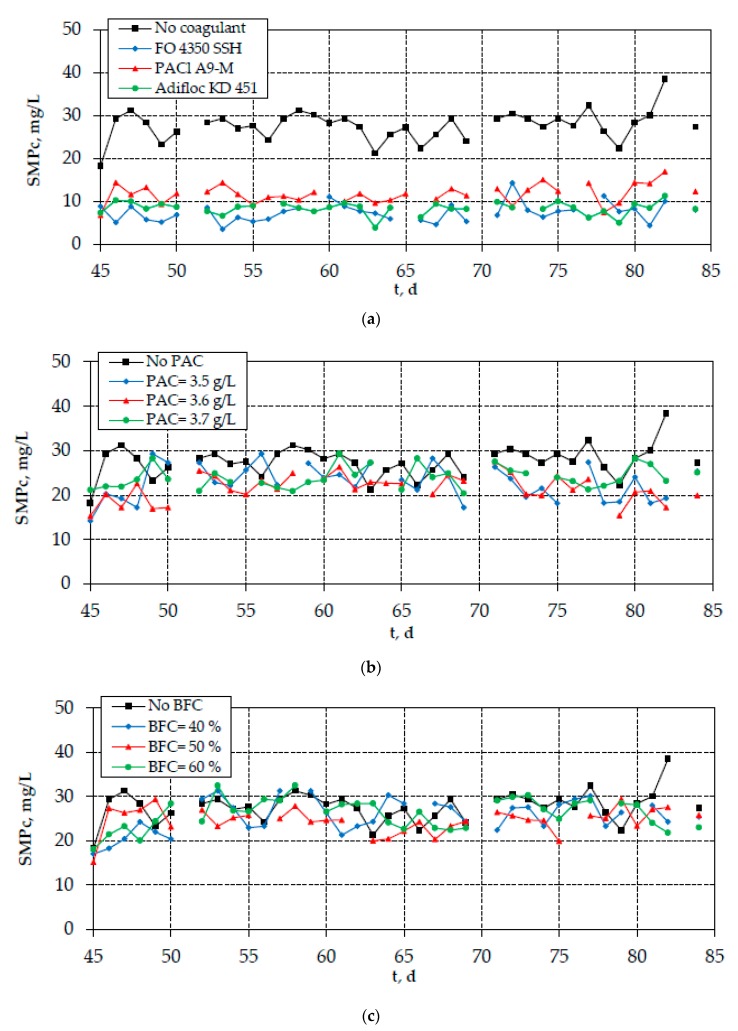
Evolution of SMP_c_ in the mixed liquor after the addition of (**a**) optimal coagulation agents FO 4350 SSH (10 mg/L), PACl A9-M (100 mg Al/L), and Adifloc KD 451 (10 mg/L); (**b**) optimal PAC concentrations (3.5–3.7 g/L); and (**c**) bio-film carriers (BFCs) (40%, 50%, and 60%).

**Figure 5 membranes-10-00042-f005:**
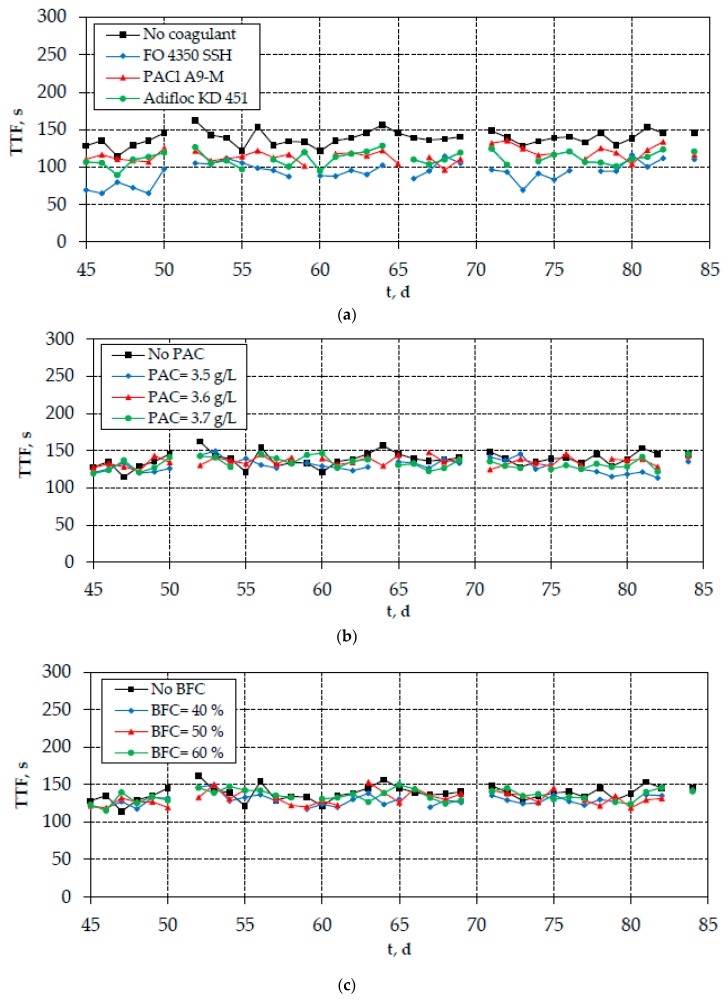
Evolution of TTF in the mixed liquor after the addition of (**a**) optimal coagulation agents FO 4350 SSH (10 mg/L), PACl A9-M (100 mg Al/L), and Adifloc KD 451 (10 mg/L); (**b**) optimal PAC concentrations (3.5–3.7 g/L), and (**c**) bio-film carriers (40%, 50%, and 60%).

**Figure 6 membranes-10-00042-f006:**
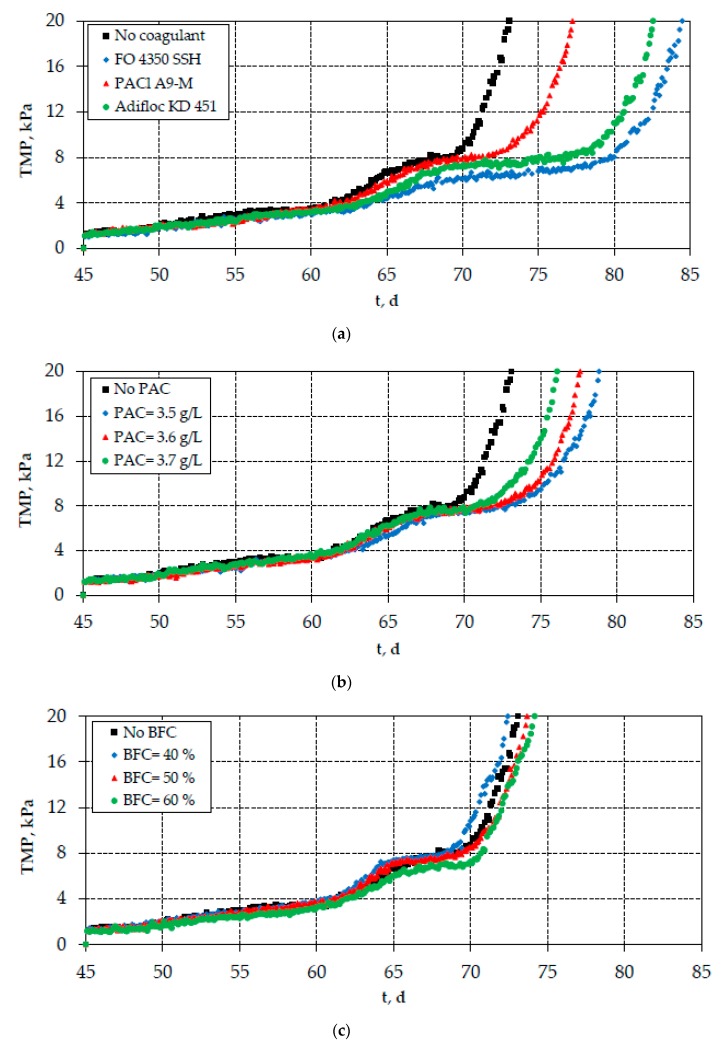
Evolution of trans-membrane pressure (TMP) in the mixed liquor after the addition of (**a**) optimal coagulation agents FO 4350 SSH (10 mg/L), PACl A9-M (100 mg Al/L), and Adifloc KD 451 (10 mg/L); (**b**) optimal PAC concentrations (3.5–3.7 g/L), and (**c**) bio-film carriers (40%, 50%, and 60%).

**Figure 7 membranes-10-00042-f007:**
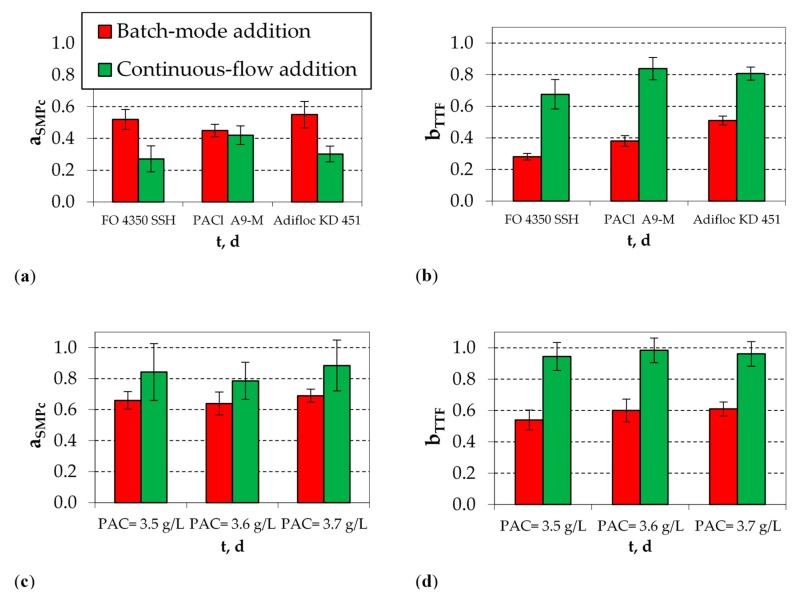
Average values of (**a**) a_SMPc_ and (**b**) b_TTF_, after batch-mode and continuous-flow addition of the optimal coagulation agents FO 4350 SSH (10 mg/L), PACl A9-M (100 mg Al/L), and Adifloc KD 451 (10 mg/L); and (**c**) a_SMPc_ and (**d**) b_TTF_, after batch-mode and continuous-flow addition of the optimal PAC concentrations (3.5, 3.6, and 3.7 g/L).

**Figure 8 membranes-10-00042-f008:**
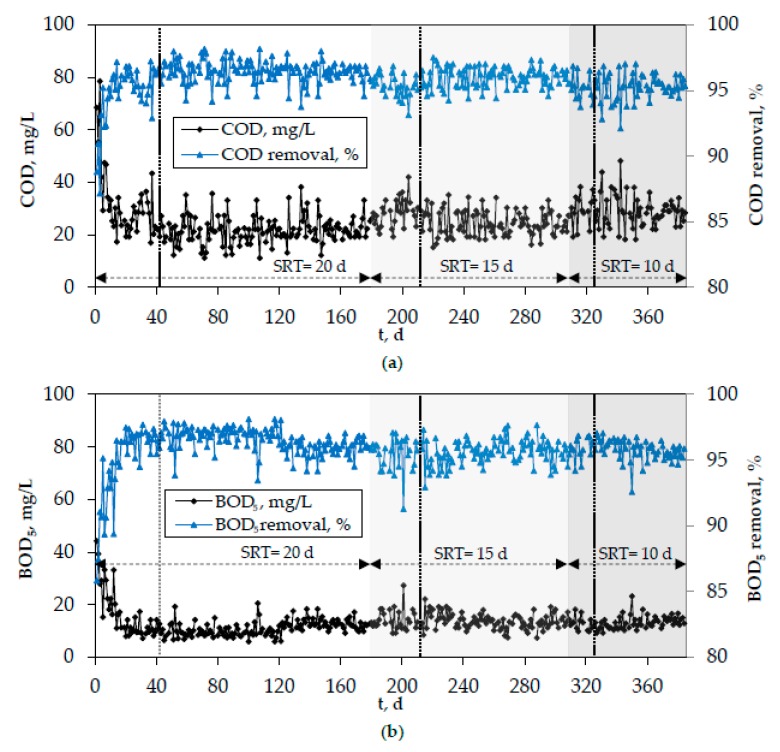
Evolution of (**a**) chemical oxygen demand (COD) και (**b**) biochemical oxygen demand (BOD)_5_ in the effluent of the lab-scale MBR and the respective removal rates.

**Figure 9 membranes-10-00042-f009:**
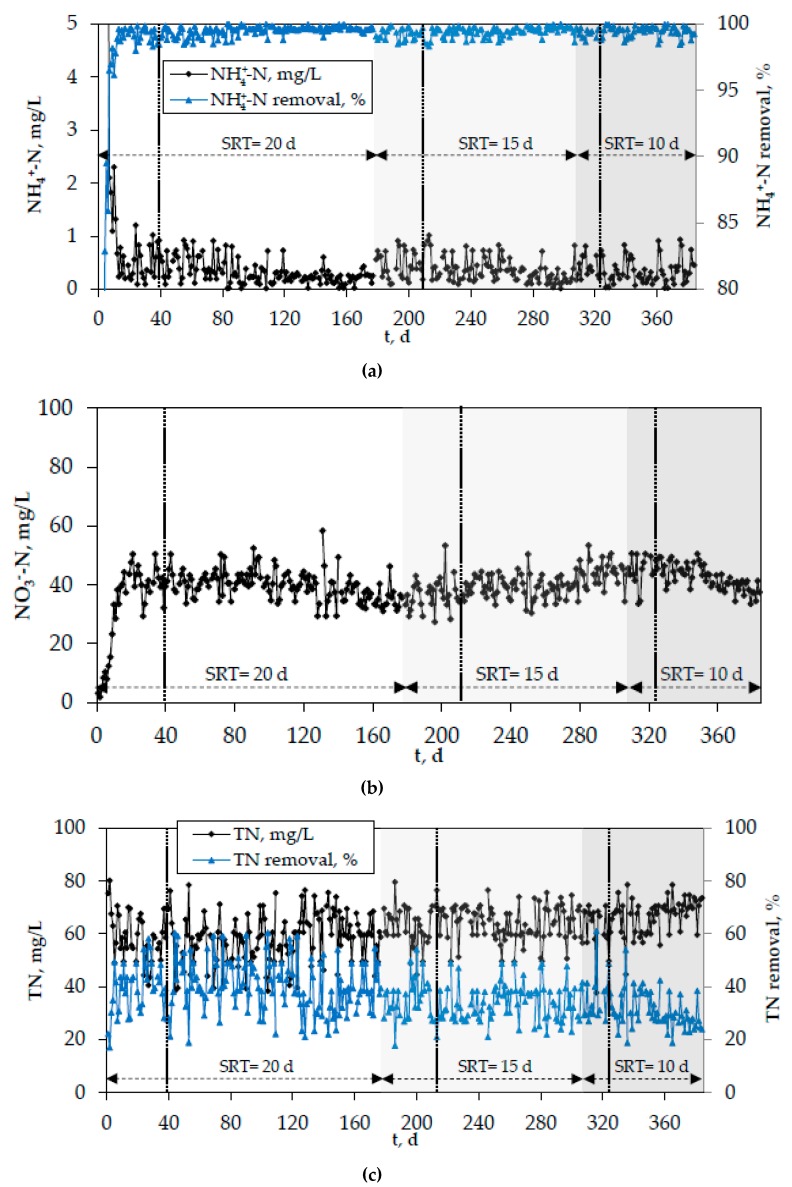
Evolution of (**a**) NH_4_^+^-N, (**b**) NO_3_^−^-N, and (**c**) total nitrogen (TN) in the effluent of the lab-scale MBR and the respective removal rates.

**Figure 10 membranes-10-00042-f010:**
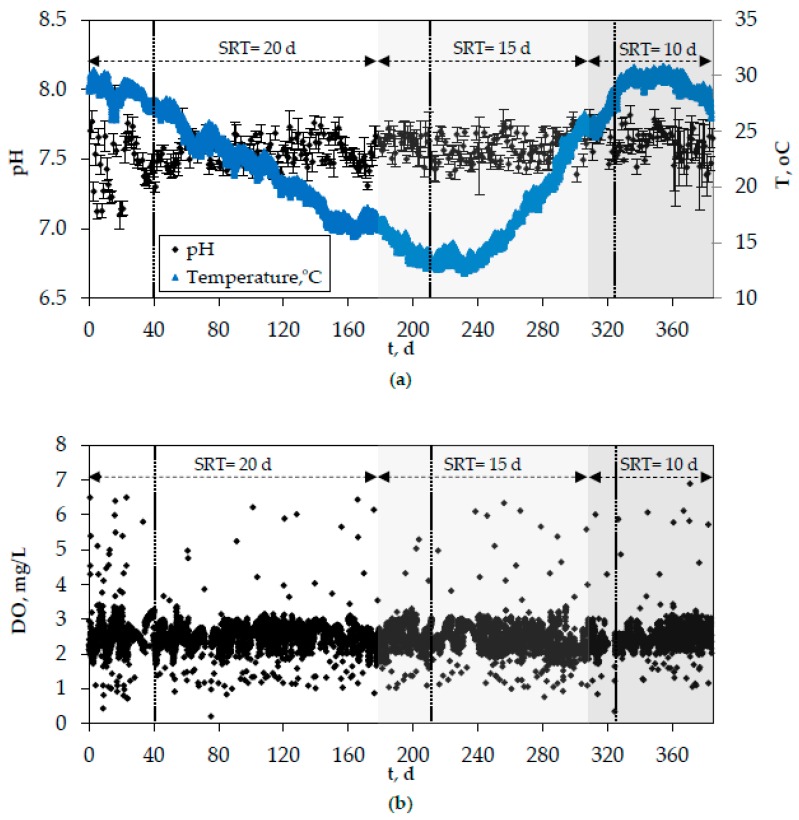
Evolution of (**a**) pH, temperature, and (**b**) dissolved oxygen (DO) concentration in the biomass of the lab-scale MBR.

**Table 1 membranes-10-00042-t001:** Composition and physico-chemical characterization of synthetic municipal wastewater. BOD, biochemical oxygen demand; COD, chemical oxygen demand; TN, total nitrogen.

Substance *	Concentration, mg/L
Peptone water	0.4
Meat extract	0.275
Urea	0.075
K_2_HPO_4_	0.07
NaCl	0.0175
CaCl_2_∙2H_2_O	0.01
MgSO_4_∙7H_2_O	0.005
**Characterization Parameter**	
COD = 611.1 ± 26.3 mg/L	
BOD_5_ = 313.1 ± 19.2 mg/L	
TN = 96.7 ± 12.8 mg/L	
NH_4_^+^ = 59.6 ± 8.1 mg/L	
NO_3_^−^ = 2.9 ± 0.5 mg/L	
pH = 7.76 ± 0.1	
Conductivity = 0.983 ± 0.04 mS/cm	

* A solution of Na_2_CO_3_ (1 M) was also periodically added in the bioreactor in order to keep pH at 7.5 ± 0.5.

**Table 2 membranes-10-00042-t002:** Coagulation/flocculation agents used in the present study.

**Simple metallic salts** FeCl_3_∙6H_2_O	
**Pre-polymerized coagulation agents** PACl-10 (5 wt.% Al) PACl-16 (8 wt.% Al) PACl-18 (9 wt.% Al) PACl A9-L (4.5 wt.% Al) 5% Polyamine of low MW ^1^ (poly-dimethylamine-co-epichlorohydrin-co-ethylenediamine) PACl A9-M (4.5 wt.% Al): 5% Polyamine of medium MW (poly-dimethylamine-co-epichlorohydrin-co-ethylenediamine)	
**Organic Polymers (cationic polyelectrolytes)**	
**Polyacrylamides**	**Polymers of diallyldimethylammonium chloride**
FO 4290 SSH (20% cationic)	Adifloc KD 550 (average M.W.)
FO 4350 SSH (25% cationic)	Adifloc KD 451 (high M.W.)
FO 4490 SSH (40% cationic)	Adifloc KD 453 (very high M.W.)
FO 4650 SSH (55% cationic)	Adifloc KD 553 (very high M.W.)

^1^ MW, molecular weight. PAC, powdered activated carbon.

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
