# Peer review of "Using Additives for Fouling Control in a Lab-Scale MBR; Comparing the Anti-Fouling Potential of Coagulants, PAC and Bio-Film Carriers"

_membranes, 2020, doi:10.3390/membranes10030042_

Round 1

Reviewer 1 Report

membranes-714541-peer-review-v1 Review:

Title: Development of a novel methodology for fouling control in MBRs with additives; comparing the anti-fouling potential of coagulants, PAC and bio-film carriers

I have finished reviewing the manuscript submitted for publication in Membranes. The overall suggestion I have is that the paper is acceptable for publication after minor revision. The manuscript contains new and valuable results and is worthy to be published.

The manuscript “Development of a novel methodology for fouling control in MBRs with additives; comparing the anti-fouling potential of coagulants, PAC and bio-film carriers” shows that the effect of different additives on the fouling propensity of a pilot-scale MBR could be very important and could increase the membrane lifetime too. Furthermore, the antifouling potential of different commonly used coagulation agents and carriers was compared. On the other hand the final effluent quality was also investigated related to the permissible discharge limits.

The paper is written in quite good English.

My specific comments and questions are as follows:

The COD concentration in the permeate depends on the feed COD content too. What was the value of COD in the feed? Was it constant or changed in the sample. Or model/synthetic wastewater was used for each experiment? How many parallel measurements were carried out? Any correlation was observed between COD and BOD5? Why BOD5 was chosen? BOD21 was also tested? It would be interesting to know than what kind of wastewaters can be treated with this technique? Are you planning to test other types/industrial waters? Why exactly 0.04 micrometer MF membrane was selected for the laboratory work? What do you think the pore size of the membrane was really 0.4 micron? Secondary membrane was formed during the filtration which could result these high rejection values?

Reviewer 2 Report

Manuscript number: membranes-714541

Title: Development of a novel methodology for fouling control in MBRs with additives; comparing the antifouling potential of coagulants, PAC and Biofilm carriers

Reviewer comments:

In this article, the authors studied a wide range of available coagulants/flocculants, PAC and biofilm carriers on the fouling propensity of a pilot MBR. The efforts to conduct so many experiments to assess the performance of the additives are highly acknowledged and appreciated. However, the authors have not brought in fundamental theories and concepts to value-add to their empirical findings. Therefore, I believe that the quality of this manuscript in its current form is not suitable for Membranes. Specific comments are as follows:

In Material and Methods, the authors should provide more detail on the analytical method used such as the COD, BOD, TN etc. As the sludge filterability test (TTF) and SMPc removal are the two key performance indicators, the methodology of these parameters should be provided instead of just providing reference for the readers. As aSMPc and bTTF are completely two different parameters, I doubt that the performance of the additives can be compared by just simple addition of these two constant. I suggest the comparison for asmpc and bTTF for different additives should be performed separately. Select the best coagulants from each category and provide possible mechanism. In short, need more explanation for example why Adifloc KD 451 ,PACL A9-M and FO 4350 SSH perform better than others, why? Can the authors provide the floc size after addition of the additives, this could explain the TTF. The explanation on the comparison between the batch mode and continuous flow mode in Figure 7 is confusing. In line 245, the authors indicated that the “filterability is more improved when the coagulants are added to biomass samples”. But from Figure 7b, the bTTF ratio is higher for continuous flow addition, the filterability is NOT improved. Similar to line 253, the SMPc is not significantly reduced when PAC is continuously added to the bioreactor. In Section 3.2, the authors provide the evolution of organic and nitrogen removal of the pilot MBR. It is unclear that the experiments were performed with or without additives. What is the COD, BOD, TN removal efficiency for different additives? Or at least for the best three coagulant/flocculant. A more detail feed/effluent quality of the MBR should be provided. For example, DOC, Ca2+, Mg2+, conductivity etc Can authors comments why total phosphorus removal is not studied in this study as this is also an important parameter for MBR? Can authors comments on how the additives affect the viability of the biomass in the reactor? Have the authors performed any toxicity test? Can authors provide cost benefit analysis or energy analysis based on the selected coagulant (FO 4350 and Adifloc KD 451)? Are they expensive? Justify.

Reviewer 3 Report

The authors submitted a paper on the comparison of different anti-fouling methods. The paper is generally well written and the topic of interest to the scientific community.

A weakness of the paper is, that both title and abstract assume more, than the trials can provide. The experimental set-up is in rather small scale and the experiments are operated on synthetic wastewater. This information is missing in the abstract. The combination of method development and pilot-scale MBR in the abstract rather gives the impression of a bigger scale and the use of real wastewater. I suggest to change the title and add this information in the abstract.

Comments in detail:

Introduction 

Some literature, i.e. on fouling quantification in the different studies should be added to validate the use of TTF, synthetic wastewater and scaling effects of fouling control by additives.

Material and Methods 

Information on tank and reactor volumes and composition of synthetic wastewater should be given. Also, a detailed description of the TTF method (which volume?) would be helpful. For the experimental design, it is not clear, if the sludge was renewed and the membrane cleaned between the different batches and how long the respective batch-trials lasted.
How was SRT adjusted in the (continuous) trials? Was there continuous sludge withdrawel (not shown in figure 1)?
Was the SMPc concentration measured in the activated sludge or in the liquid phase of the activated sludge? Please clarify or correct in line 182.

Results

Batch mode operation and definition of parameters a and b should be included in Material and Methods.
Figure 2: Error bars and an optical separation between different additives would be helpful to understand the results.

The understanding of the results of the continuous-flow experiments needs some information about the experimental set-up. Figures 4-6 shows results on different additives, PAC and BFC over the same time scale. Were there 10 reactors operated in parallel?

Could the authors please provide the MLSS concentrations for all results presented, as it is known from literature that TTF also depends on MLSS concentration?

Round 2

Reviewer 2 Report

The authors have addressed the comments raised by the reviewers. With the revised version, I strongly suggest this article to be published in Membrane.

Author Response

We would like to thank Reviewer 2 very much for the useful remarks.

Reviewer 3 Report

The authors have resubmitted the paper with many amendmends in the text passages. Expecially the change of title and amendmends of the abstract paragraph are appreciated and improve the paper.

Most of my previous comments regarded the description of the experimental design and its implications on results and discussion. The authors tried to clarify many questions, but to my point of view, it is still not possible to fully understand the experimental approach. In detail:

1) The authors have added in line 185 the procedure for TTF, i.e. they write that different filter papers were used. Is this really true? In this case, the results would not be comparable.

2) Sludge pretreatment for SMP analyses is given by centrifugion. Please, add details.

3) The authors added, that sludge was replaced in the continuous trials. Was the membrane module also replaced, or was ist cleaned? How did initial flux change?

4) Definition of SRT and procedure for adjustment needs to be described more clearly. Line 157/158 now read: "In order to achieve the desirable SRT in both phases, appropriate amount of sludge was daily removed from the bioreactor." What is an appropriate amount? How can SRT be controlled by sludge withdrawal and MLSS at the same time be kept constant (lines 132/133)? If so, other parameters probably have changed (sludge load, F/M ratio), which also might influence fouling and should be discussed. Sludge withdrawal is not shown in figure 1b.

5) Sludge load and F/M ratio should be mentioned.

6) Batch tests: by the now described procedure it becomes clear, that batch trials are undertaken outside the reactor. The authors should add sample volume and comtact time of additives before analyses.

7) Batch results: The authors state that figure 2 will be difficult to read with error bars. For discussion of the results, this is a vital information, though, especially as many additives show different behaviour at different SRT.

8) Results continuous trials: although the authors have given further information in Materials and Methods, it is still difficult to understand figures 4 - 6. In what order and time frame were the different experiments performend? For 2x 20d and new sludge, it must be at least 40d control, 40 d FO, 3x 40d PAC, 40 d Adifloc, 3x 40d BFC --> which amounts to 400 d while the period obviously only lasted 317 d. Also, it is not clear, why the time axis starts at 45 days. It would be very helpful, if the results were given in chronological order at least for figures 4 and 5.

9) Also in the figure regarding removal efficiency, time axis is difficult to understand. In lines 320/321 it reads: "In these figures, the period of SRT= 20 days refers to the initial period of the second phase when the 321
optimal coagulants were continuously added." This period is 180 days.

Furthermore, some minor corrections in the newly written passages are necessary:

1) Table 1 needs to be devided, as the third column is not a function of the first column.

2) "as" TTF, line 188

3) Hach-Lange, line 198 (spelling and the numbers of the test kits used should be given)

Round 3

Reviewer 3 Report

The authors have included all relevant comments.